# Characterizing the dynamics of the rumen microbiota, its metabolites, and blood metabolites across reproductive stages in Small-tailed Han sheep

Yuzhu Sha,[1] Xiu Liu,[1] Xiaoning Pu,[1] Yanyu He,[2] Jiqing Wang,[1] Shengguo Zhao,[1] Pengyang Shao,[1] Fanxiong Wang,[1] Zhuanhui Xie,[1] Xiaowei Chen,[1] Wenxin Yang[1]

**ABSTRACT** Different reproductive stages of mammals involve complex biological processes, and the intestinal microbiota, as an endocrine organ or an "invisible organ," is involved in the regulation of hormone levels, immune function, and metabolism. However, the effects of the rumen microbiota, its metabolites, and blood metabolites on the reproductive performance of ruminants remain unclear. This study revealed that the *Prevotella* abundance increased significantly during pregnancy ($P < 0.01$); the *Fibrobacter* abundance increased significantly during lactation ($P < 0.05$); and rumen microbial carbohydrate metabolism, glucose biosynthesis, and metabolic functions were significantly enriched during pregnancy ($P < 0.05$). Microbial metabolic profile analysis showed that the differentially abundant microbial metabolites during pregnancy and lactation were mainly enriched in the biosynthesis of ubiquinone and other terpenoid quinones, and there was a certain correlation with the microbiota. Among them, sapindoside A was increased during pregnancy, nicotinamide riboside and β-cryptoxanthin were reduced during pregnancy, and L-tryptophan was significantly increased during lactation. In addition, the volatile fatty acid levels in lactation were significantly higher than those in non-pregnancy and pregnancy ($P < 0.05$), and the $NH_3$-N content during pregnancy was significantly higher than that during lactation and non-pregnancy ($P < 0.05$). Moreover, there were differences in the serum metabolite levels at different reproductive stages, and similar metabolites existed when comparing the rumen metabolites, which were mainly enriched in arachidonic acid metabolism, vitamin B6 metabolism, and ABC transporter protein, resulting in significantly higher serum IgA and IgM levels during lactation than during non-pregnancy and pregnancy ($P < 0.05$).

**IMPORTANCE** Our study illustrates the succession of the rumen microbiota and its metabolites in Small-tailed Han sheep at different reproductive stages. Among them, Firmicutes and *Prevotella*, which are related to energy metabolism, increased in abundance during pregnancy, while *Fibrobacter*, a fiber-degrading bacterium, increased in abundance during lactation. At the same time, the microbial metabolic profile and serum metabolic profile characteristics of different reproductive stages were revealed, and some functional pathways and metabolites related to energy and immunity were found. This study provides a reference for the health management of ruminants during non-pregnancy, pregnancy, and lactation.

**KEYWORDS** reproductive, rumen, microbiota, metabolites

Address correspondence to Xiu Liu, liuxiu@gsau.edu.cn.

The authors declare no conflict of interest.

See the funding table on p. 15.

S mall-tailed Han sheep has the characteristics of fast growth, strong fecundity, high yield of multiple births, perennial estrus, and strong adaptability and are often selected as excellent breeding ewes. Their reproductive performance is an important

index to measure production performance, which determines the reproductive rate and number of lambs. The health status and feeding management of ewes in different reproductive stages are important factors affecting their reproductive performance. During non-pregnancy, pregnancy, and lactation, the mother will undergo complex biological processes, and a series of changes will take place in the levels of hormones, immunity, and metabolism. These physiological changes play an important role in the health of the mother and offspring (1). At present, studies on sheep at different reproductive stages mainly focus on blood physiology and biochemistry, hormone indices, and some metabolic levels (2, 3), while it is not clear whether the maternal intestinal microbiota and its metabolites are involved in the regulation of physiological metabolic processes in different reproductive stages of Small-tailed Han sheep.

The gastrointestinal tract of ruminants is inhabited by a large number of microbiota, including bacteria, archaea, fungi, and protozoa, which can digest protein, starch, and cellulose and produce volatile fatty acids (VFAs), microbiota proteins, ammonia nitrogen, and other metabolites (4). These metabolites not only provide energy for the body, but also play a key role in the occurrence of disease, physiological regulation, and health of the host (5). The gastrointestinal tract and its accompanying intestinal microbiome constitute the main site of immune and endocrine regulation (6), affecting distal organs and pathways, and are considered to be a fully functional endocrine organ (7) involved in the physiological regulation of the body. With the development of omics technology, researchers continue to deepen the study of the gastrointestinal microbiota, and it has been found that the intestinal microbiota are closely related to the reproductive performance of animals (8). The intestinal microbiota affect maternal metabolic homeostasis and fetal growth and development through VFAs, inflammatory factors, bile acid metabolism, and hormones (9, 10). Studies have reported that the gut microbiota are associated with the production of reproductive hormones (11) and interact with the microbiota to affect the metabolism and immunity of the host (5). Studies have found that the microbiota affect every stage of female reproduction, including follicle and oocyte maturation, fertilization, embryo migration, and implantation, as well as throughout pregnancy and even during delivery, and that correcting an abnormal microbiota may improve reproductive outcomes (12). Studies have found that there were some differences in the posterior gut microbiota of goats at different reproductive stages, and significant correlations were found between the microbiota and reproductive hormones and immune indicators (13). These studies indicate that the gut microbiota play a role in the regulation of reproductive processes. Intestinal microbiota changes at different stages of pregnancy, especially in the third trimester, resulting in changes in metabolic, immune, and hormone levels, which are beneficial to maternal health during pregnancy and fetal growth and development (8). In turn, dramatic changes in hormone levels also affect the composition and function of the microbes and may be accompanied by unique inflammatory and immune changes. Increased progesterone, proinflammatory factors, and decreased immune activity in prenatal serum alter gut function and the microbiota composition (8, 14). The fecal microbial diversity of sows during late pregnancy and early postpartum revealed that compared with sows with small litter size, sows with large litter size had a lower microbial richness in the late pregnancy stages, while sows with large litter size had a higher microbial richness in the early postpartum stages (15). Therefore, it is important to understand the effects of the rumen microbiota and its metabolites on host metabolism in ewes at different reproductive stages for healthy feeding and management of breeding ewes. In this study, the interaction of the rumen microbiota and its metabolites with blood metabolites in different reproductive stages (non-pregnancy, pregnancy, and lactation) of Small-tailed Han sheep was analyzed to reveal the succession and trends in the changes in the rumen microbiota, its metabolites, and blood metabolites in different reproductive stages, which will provide a basis for the health management of ruminants during non-pregnancy, pregnancy, and lactation.

## RESULTS

### Succession of the rumen microbiota in different reproductive stages

16S rRNA sequencing was performed on the rumen microbiota at different reproductive stages, and a total of 1,244 operational taxonomic units (OTUs) were obtained (Fig. S1A). According to the dilution curve (Fig. S1B), the coverage of the sequencing data was saturated, which, therefore, represents the real situation of the sample. Principal co-ordinates analysis (PCoA) showed that there were certain differences in the rumen microbiota at different reproductive stages (Fig. S1C). The alpha diversity index analysis showed that there was no significant difference in the diversity indices of the three stages (Table S1), among which the abundance-based coverage estimator (ACE) index and Chao1 index were higher in the pregnancy stages than in the lactation and non-pregnancy stages, but the difference was not significant ($P > 0.05$).

A total of 18 phyla, 26 classes, 38 orders, 64 families, 151 genera, and 175 species were identified at different taxonomic levels. At the phylum level, Bacteroidetes and Firmicutes were the dominant phyla (Fig. 1A), and Fibrobacteres in lactation was significantly higher than that in non-pregnancy and pregnancy stages ($P < 0.05$). Spirochaetes were significantly higher in lactation than in pregnancy ($P < 0.05$) (Table S2). At the genus level (Fig. 1B; Table S3), *Prevotella*, *Succiniclasticum*, and *Rikenellaceae RC9* were the dominant genus. *Prevotella* was significantly higher in pregnancy than in lactation ($P < 0.01$). *Ruminococcus* was significantly higher during pregnancy than during non-pregnancy ($P < 0.05$). The level of *Fibrobacter* in non-pregnancy stages was significantly lower than that in lactation stages ($P < 0.05$). The levels of *Helicobacter* and *Erysipelotrichaceae UCG009* in lactation were significantly lower than those in non-pregnancy stages ($P < 0.05$). The levels of *Moryella*, *Lachnoclostridium*, and *Fibrobacter* in lactation were significantly higher than those in pregnancy stages ($P < 0.01$), while *Treponema* in lactation was significantly higher than that in pregnancy stages ($P < 0.05$). Line discriminant analysis effect size (LEfSe) analysis of samples between groups showed three different biomarkers during pregnancy, with *Prevotella* being the main one (Fig. 2).

Kyoto Encyclopedia of Genes and Genomes (KEGG) functional analysis (Table 1) showed that carbohydrate metabolism, glycan biosynthesis, and metabolism were significantly higher in pregnancy than in lactation ($P < 0.05$); the metabolism of terpenoids and polyketides was significantly higher in lactation than in pregnancy ($P < 0.05$); and digestive system function was significantly higher in pregnancy than in

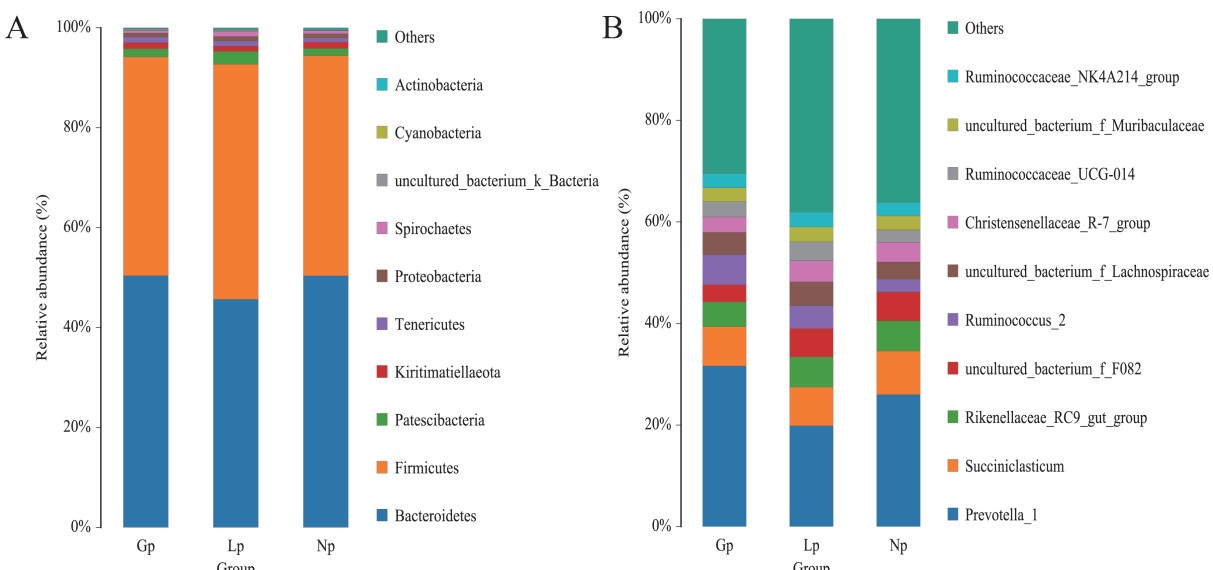

**FIG 1** Species composition analysis. (A) Species composition at the phylum level. (B) Species composition at the genus level. NP, non-pregnancy; Gp, pregnancy; LP, lactation.

**TABLE 1** Functional prediction analysis of KEGG and COG at different reproductive stages (%)[a]

| KEGG pathways | Non-pregnancy | Pregnancy | Lactation | P value |
|---|---|---|---|---|
| Carbohydrate metabolism | –[b] | 9.725 | 9.539 | 0.012 |
| Metabolism of terpenoids and polyketides | – | 1.055 | 1.082 | 0.039 |
| Glycan biosynthesis and metabolism | – | 1.652 | 1.574 | 0.039 |
| Digestive system | 0.098 | 0.116 | – | 0.011 |
| Cancers: Overview | 0.060 | 0.057 | – | 0.029 |
| Metabolism of terpenoids and polyketides | 1.064 | – | 1.082 | 0.018 |
| Endocrine and metabolic diseases | 0.200 | – | 0.203 | 0.020 |
| **COG pathways** | **Non-pregnancy** | **Pregnancy** | **Lactation** | **P value** |
| Nucleotide transport and metabolism | – | 3.911 | 3.810 | 0.012 |
| Inorganic ion transport and metabolism | – | 4.311 | 4.396 | 0.038 |
| Cell motility | – | 0.801 | 0.922 | 0.005 |
| Inorganic ion transport and metabolism | 4.383 | 4.311 | – | 0.025 |
| Intracellular trafficking, secretion, and vesicular transport | 1.880 | 1.836 | – | 0.047 |

[a]Functional analysis is a pairwise comparison, and $P < 0.05$ represents a significant difference.
[b]– means that the function is not commented to this group and is compared between the other two groups.

non-pregnancy ($P < 0.05$). In the Clusters of Orthologous Groups of proteins (COG), gene family, cell motility, inorganic ion transport, and metabolism were significantly higher in lactation than in pregnancy ($P < 0.05$), and nucleotide transport and metabolism were significantly higher in pregnancy than in lactation ($P < 0.01$). Inorganic ion transport and metabolism and intracellular trafficking, secretion, and vesicular transport pathways in non-pregnancy were significantly higher than those in pregnancy ($P < 0.05$).

## Changes in the rumen microbial metabolites in different reproductive stages

Principal component analysis showed that there were certain differences in metabolites of the rumen microbiota in different reproductive stages (Fig. 3A). With fold change (FC) >1, $P$ value <0.05, and variable important in projection (VIP) >1 as the screening criteria (Fig. 3B), 263 differential metabolites were found between non-pregnancy and pregnancy; 470 differential metabolites, between pregnancy and lactation; and 462 differential metabolites, between non-pregnancy and lactation, among which 67 common differential metabolites were found in three reproductive stages. Further screening of the top 10 differential metabolites was performed (Fig. S2), and sapindoside A, cytidine diphosphate diacylglycerol (CDP-DG) [18:1(11Z)/16:0], ampicillin, and nicotinamide riboside were found between non-pregnancy and pregnancy. Between pregnancy and lactation, CDP-DG [16:0/22:6(4Z,7Z,10Z,13Z,16Z,19Z)], phosphatidic acid (PA) [12:0/16:1(9Z)], L-tryptophan, butyrylcarnitine, etc., were identified. Between non-pregnancy and lactation, CDP-DG [16:0/22:6(4Z,7Z,10Z,13Z,16Z,19Z)],

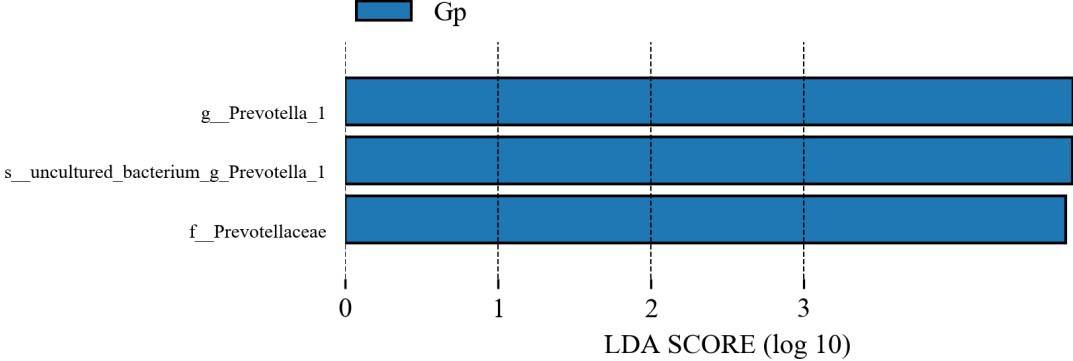

**FIG 2** LDA value distribution histogram. Linear Discriminant Analysis (LDA) value >4, and the length of the bar chart represents the influence of different species. Gp, pregnancy.

CDP-DG [18:1(11Z)/16:0], PE [18:1(11Z)/16:0], etc., were found. K-means cluster analysis of differential metabolites showed that there were 18 major clusters (Fig. 3C), among which phosphatidylcholine (PC) [18:2(9Z, 12Z)/16:0] in Cluster 7 was significantly increased during pregnancy. In Cluster 11, PC [18:0/18:1(9Z)] and PC [22:4(7Z,10Z,13Z,16Z)/P-18:0] increased during pregnancy and lactation, reaching a peak in lactation. In Cluster 17, PE [18:1(11Z)/19:0] and PC (16:0/18:0) reached the highest levels during pregnancy. The levels of 1-stearoyl-2-hydroxy-sn-glycero-3-phosphocholine and β-cryptoxanthin in Cluster 13 were reduced during pregnancy and lactation, and decreased to their lowest levels in lactation (Fig. S3).

KEGG functional annotation analysis of the differential metabolites revealed (Fig. 4) that the differential metabolites of the three stages were mainly annotated in amino acid metabolism and biosynthesis of other secondary metabolites. The enrichment network analysis showed that between non-pregnancy and pregnancy, the metabolites were mainly enriched in ubiquinone and other terpenoid-quinone biosynthesis and indole alkaloid biosynthesis. Between pregnancy and lactation, the metabolites were mainly enriched in ubiquinone and other terpenoid-quinone biosynthesis and glycine, serine, and threonine metabolism. Between non-pregnancy and lactation, the metabolites were mainly enriched in ubiquinone and other terpenoid-quinone biosynthesis; glycine, serine, and threonine metabolism; and alpha-linolenic acid metabolism.

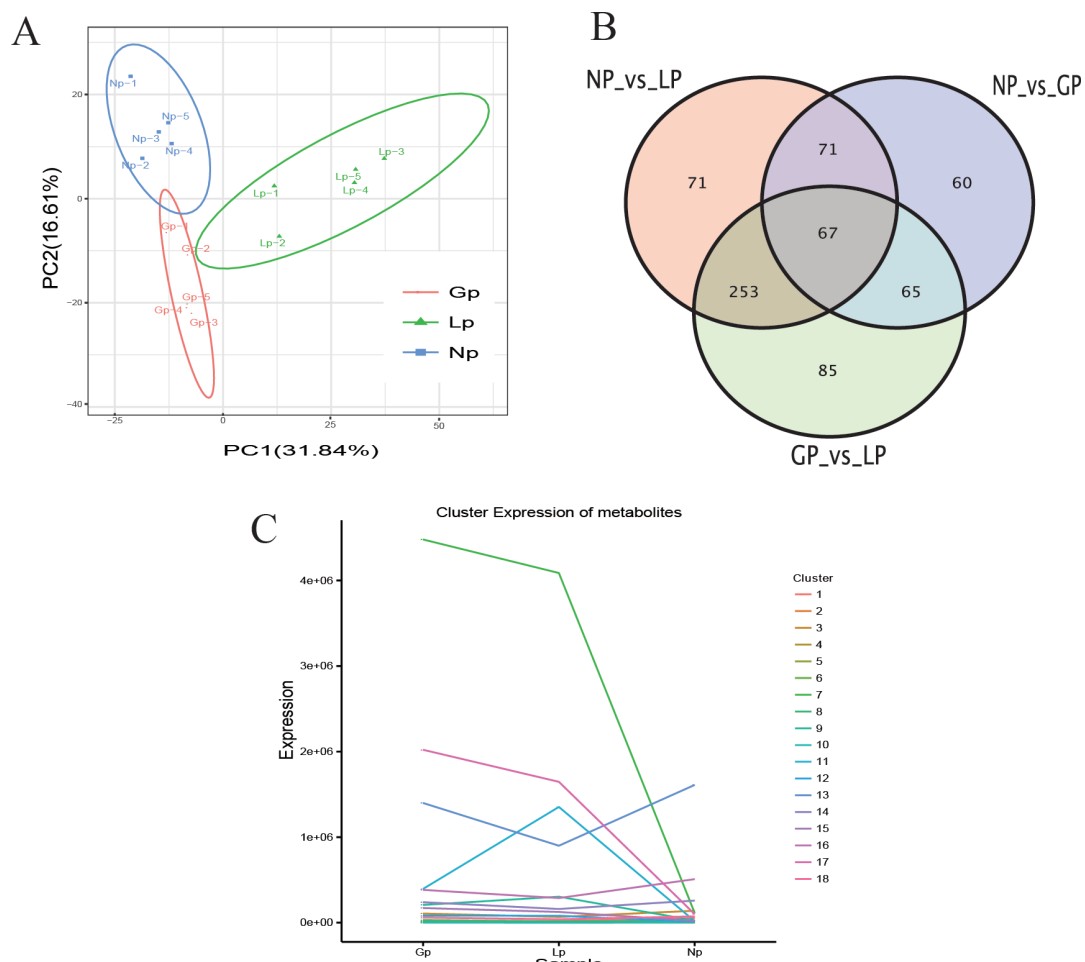

**FIG 3** Statistical map of the microbial differential metabolites. (A) PCA; (B) Venn diagram; (C) K-means cluster analysis. NP, non-pregnancy; Gp, pregnancy; LP, lactation.

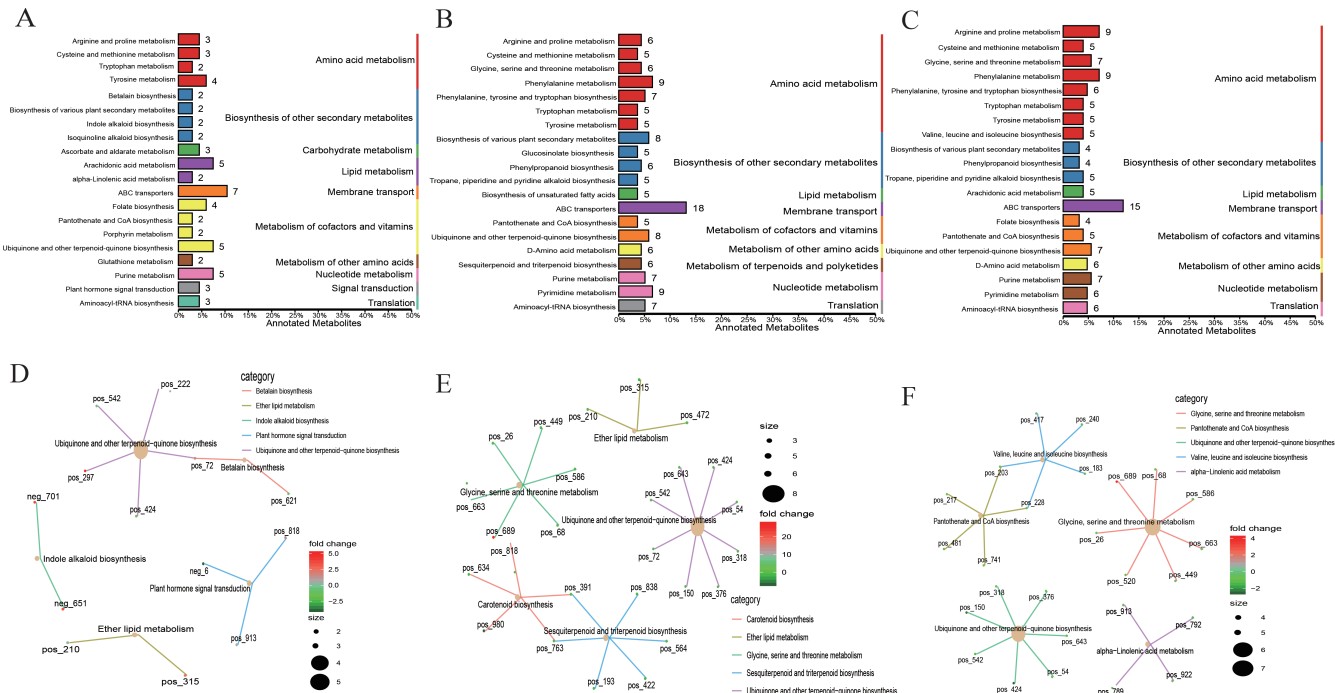

**FIG 4** KEGG functional analysis of the microbial metabolites. (A–C) KEGG annotation of the classification diagram; (D–F) KEGG enrichment network of differential metabolites. Note: A and D, non-pregnancy vs pregnancy; B and E, pregnancy vs lactation; C and F, non-pregnancy vs lactation.

## Rumen fermentation function in different reproductive stages

As shown in Table 2, there were differences in the rumen VFA content in different reproductive stages of Small-tailed Han sheep, and the total VFA content in lactation was significantly higher than that in non-pregnancy and pregnancy ($P < 0.05$), among which the acetate content in lactation was significantly higher than that in non-pregnancy and pregnancy ($P < 0.05$), and the ratio of acetate to propionate (A: $P$) and the acetate concentration showed the same trend, but the butyrate concentration had no significant difference ($P > 0.05$). The $NH_3$-N content at different reproductive stages was significantly higher in the pregnancy stage than in the lactation and non-pregnancy stages ($P < 0.05$).

## Combined analysis of the rumen microbiome and metabolome

According to Procrustes analysis, differences were found between the rumen microbiota and its metabolites in different reproduction stages (Fig. S4). The metabolite data were divided into different metabolite modules by weighted correlation network analysis dimension reduction analysis, and correlation analysis with phylum level bacteria was

**TABLE 2** Ruminal VFA and $NH_3$-N contents at different reproductive stages[a]

| Parameter | Non-pregnancy | Pregnancy | Lactation |
|---|---|---|---|
| Acetate (mmol/L） | $24.89 \pm 3.16^c$ | $30.34 \pm 0.64^b$ | $36.94 \pm 2.92^a$ |
| Propionate (mmol/L） | $8.15 \pm 2.83$ | $8.03 \pm 2.56$ | $5.22 \pm 0.02$ |
| A/P | $3.43 \pm 1.74^b$ | $4.09 \pm 1.46^b$ | $7.08 \pm 0.58^a$ |
| Butyrate (mmol/L） | $4.29 \pm 0.67$ | $4.37 \pm 0.61$ | $5.21 \pm 0.82$ |
| Valerate (mmol/L） | $0.72 \pm 0.03$ | $0.83 \pm 0.16$ | $0.70 \pm 0.10$ |
| Isovalerate (mmol/L） | $0.79 \pm 0.06$ | $0.65 \pm 0.03$ | $0.78 \pm 0.17$ |
| Isobutyrate (mmol/L） | $0.67 \pm 0.03$ | $0.58 \pm 0.03$ | $0.62 \pm 0.00$ |
| Total VFA (mmol/L） | $39.51 \pm 2.96^b$ | $44.82 \pm 3.26^b$ | $49.47 \pm 2.62^a$ |
| $NH_3$-N (mg/dL) | $4.09 \pm 0.68^b$ | $5.93 \pm 1.04^a$ | $3.00 \pm 0.17^c$ |

[a]Entries marked with different lowercase letters (a, b, c) indicate significant differences ($P < 0.05$), and the same letters or no letters indicate non-significant differences ($P > 0.05$).

carried out (Fig. S4). Seven metabolite modules in the non-pregnancy and pregnancy groups were significantly correlated with Firmicutes, Synergistetes, and Cyanobacteria ($P$ < 0.05); four modules in the pregnancy and lactation groups were significantly correlated with Firmicutes, Fibrobacteres, and Lentisphaerae ($P$ < 0.05); and five modules in non-pregnancy and lactation groups were significantly correlated with Bacteroidetes, Patescibacteria, Atribacteria, and Fibrobacteres ($P$ < 0.05).

Correlation analysis of differential metabolites and differential microbiota (genus level) showed that there was a strong correlation at different reproductive stages (Fig. S5). As correlation coefficient |CC| >0.8 and correlation coefficient $P$ value (CCP) <0.05 were the standards, the top 30 differential metabolites/differential microbiota based on frequency were identified by screening to generate a correlation network diagram (Fig. 5). Among the non-pregnancy and pregnancy stages, 19 different metabolites were found to be correlated with 11 different microbiota, among which carnitine and nicotinamide riboside were significantly positively correlated with *Papillibacter* and *Ruminococcaceae*. During pregnancy and lactation, it was found that menaquinone was positively correlated with *Prevotella*, and 2-undecanone was positively correlated with *Anaerovibrio*. Among the non-pregnancy and lactation stages, there was a significant positive correlation between PC (16:0/0:0) and *Helicobacter*.

## Blood metabolites in different reproductive stages

At different reproductive stages, there were some differences in blood metabolites (Fig. 6). FC >1, $P$ value <0.05, and VIP >1 were used as the screening criteria, and differential metabolite analysis revealed 264 differential metabolites between non-pregnancy and pregnancy, 221 differential metabolites between pregnancy and lactation, and 180 differential metabolites between non-pregnancy and lactation, among which 11 common differential metabolites were found in the three reproductive stages. The differential metabolites with the top 10 differential multiples were further screened (Fig. S6). It was found that 25-hydroxycholesterol, oleanolic acid, nicotinuric acid, ethyl-4-hydroxybenzoate, and ubiquinone 4 metabolites existed between non-pregnancy and pregnancy. Between pregnancy and lactation, E-linalool oxide, ethylmalonic acid, etc., were found. Between non-pregnancy and lactation, metabolites such as 2-amino-2-methyl-1, 3-propanediol, nicotinuric acid,

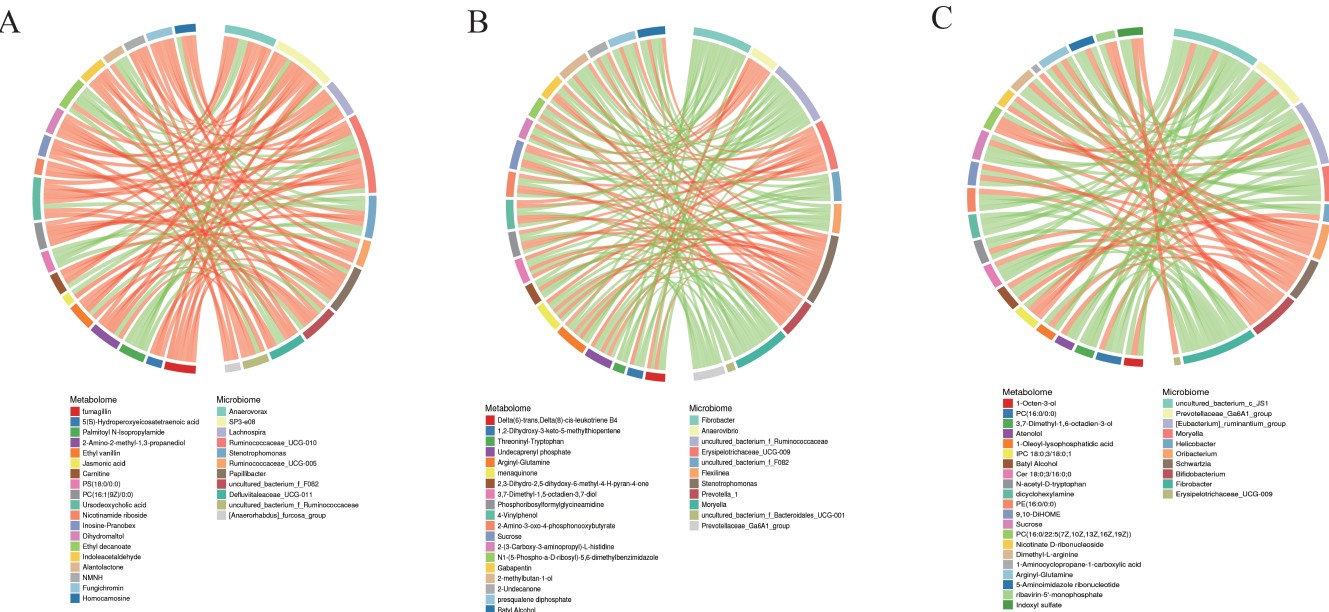

**FIG 5** Correlation network diagram of differential metabolites/differential microbiota. Note: Red lines indicate a positive correlation, and green lines indicate a negative correlation. Note: A, non-pregnancy vs pregnancy; B, pregnancy vs lactation; C, non-pregnancy vs lactation.

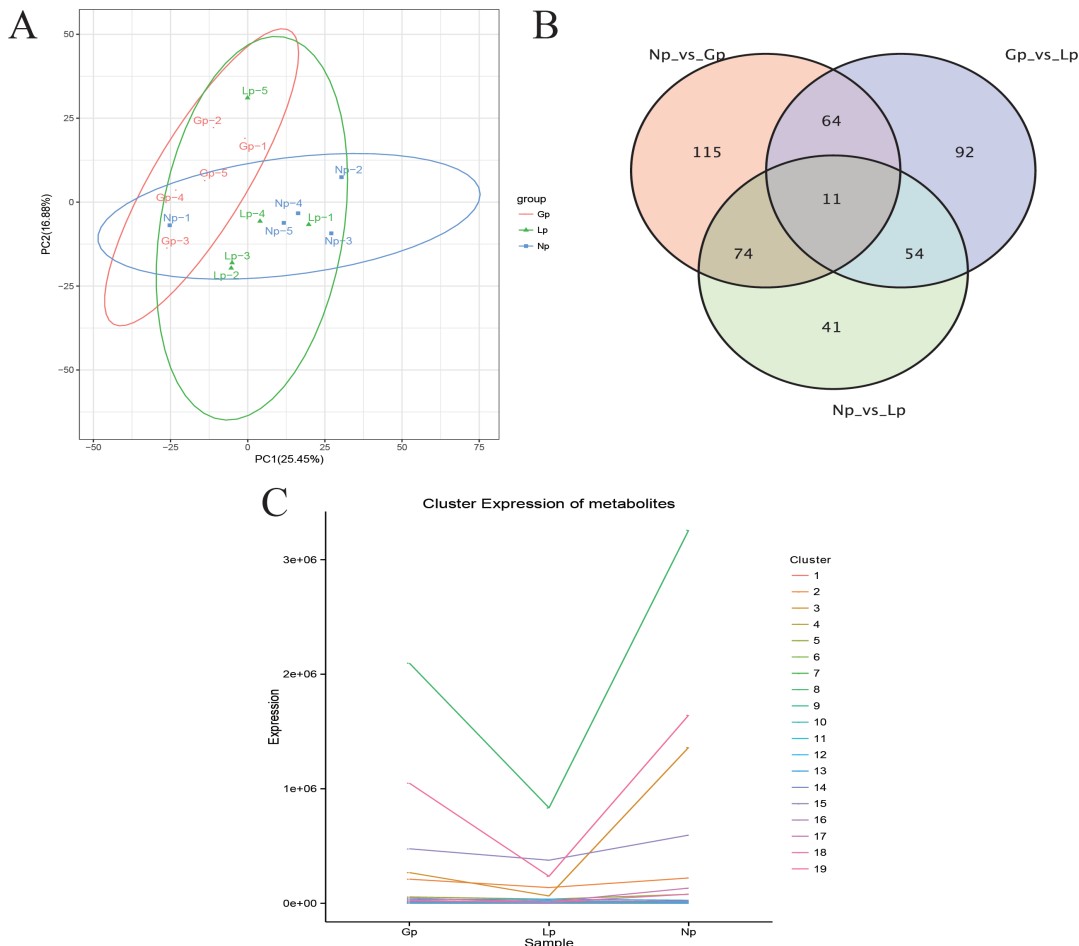

**FIG 6** Statistical map of the blood differential metabolites. (A) PCA; (B) Venn diagram; (C) K-means cluster analysis. NP, non-pregnancy; Gp, pregnancy; LP, lactation.

and PC [18:3(9Z,12Z,15Z)/22:2(13Z,16Z)] were revealed. Further K-means clustering analysis of differential metabolites showed that there were 19 major clusters, among which Cluster 8 {PC[18:2(9Z,12Z)/16:0], PC[22:4(7Z, 10Z,13Z,16Z)/P-18:0]}, Cluster 18 {PC[18:4(6Z,9Z,12Z,15Z)/20:1(11Z)], PC[18:4(6Z, 9Z,12Z,15Z)/20:2(11Z,14Z)]}, and Cluster 3 {PE[18:1(11Z)/19:0]} all decreased during pregnancy and lactation, reaching the lowest levels in lactation (Fig. S7).

Further KEGG functional analysis showed that the differential metabolites were mainly annotated in amino acid metabolism, biosynthesis of other secondary metabolites, carbohydrate metabolism, and lipid metabolism (Fig. 7). Functional enrichment network analysis showed that the differential metabolites between non-pregnancy and pregnancy were mainly enrichment in arachidonic acid metabolism and histidine metabolism. The differential metabolites between pregnancy and lactation were mainly enriched in vitamin B6 metabolism, tryptophan metabolism, and biotin metabolism, while the differential metabolites between non-pregnancy and lactation were mainly enriched in ABC transporters.

## Blood immunity in different reproductive stages

As shown in Fig. 8, there were differences in the blood immune indices of ewes of Small-tailed Han sheep at different reproductive stages, and the contents of immunoglobulin A (IgA) and immunoglobulin M (IgM) in the lactation stage were significantly higher than those in the non-pregnancy and pregnancy stages ($P < 0.05$), and the

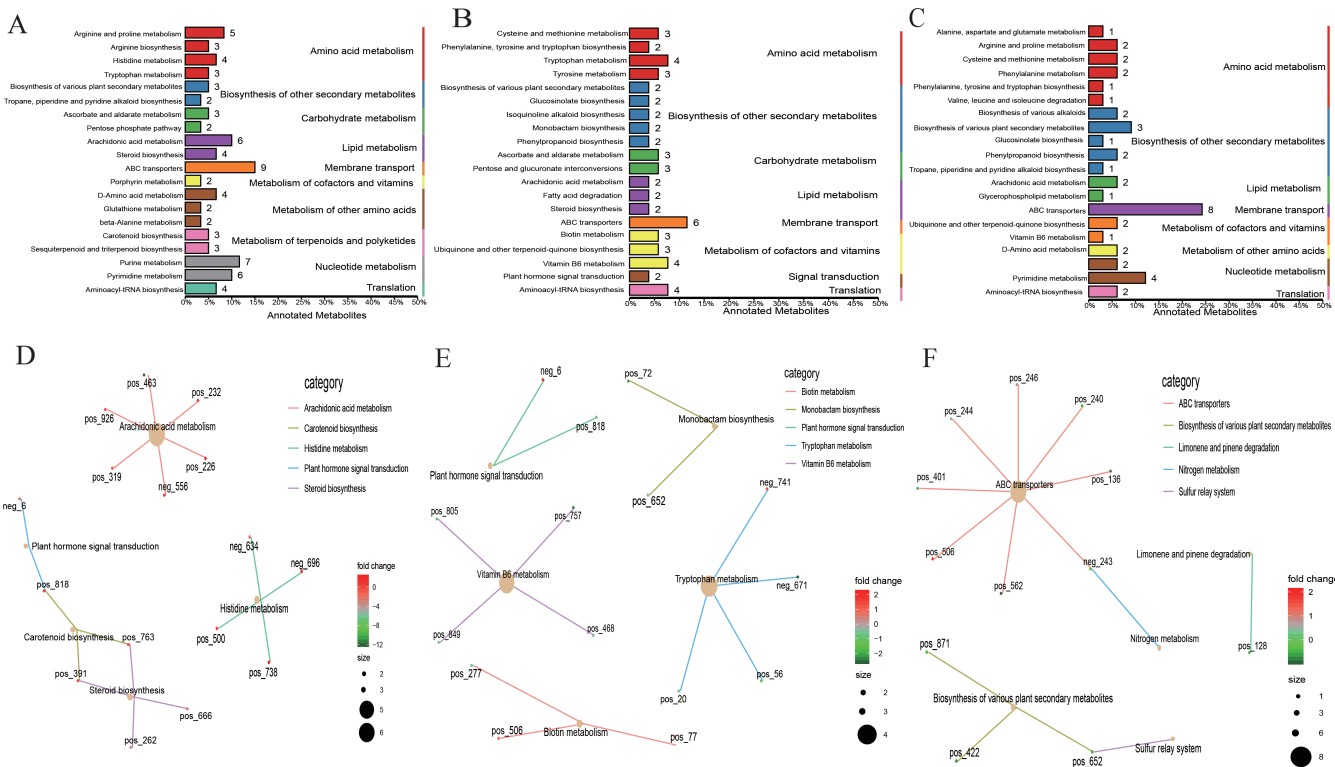

**FIG 7** KEGG functional analysis of differential blood metabolites. (A–C) KEGG annotates the classification diagram; (D–F) KEGG enrichment network of differential metabolites. Note: A and D, non-pregnancy vs pregnancy; B and E, pregnancy vs lactation; C and F, non-pregnancy vs lactation.

immunoglobulin G (IgG) content in the lactation stage was significantly higher than that in the pregnancy stages ($P < 0.05$).

## Comparative analysis of the microbial metabolites and blood metabolites

The screening criterion used was |log2FC| >2, and the differential microbial metabolites and serum metabolites were screened for comparative analysis. There were seven common differential metabolites in non-pregnancy and pregnancy, three common differential metabolites in pregnancy and lactation, and 16 common differential metabolites in non-pregnancy and lactation. Only six different metabolites in the three comparison groups were found to have the same content trend in the rumen and serum. The Human Metabolome Database (HMDB) functional classification of the different common metabolites of the three groups was further analyzed (Fig. 9). In pregnancy, the contents of nicotinamide riboside, salicylic acid, and ethyl 9-decenoate were reduced in the rumen and serum, and sapindoside A was increased in the rumen and decreased in the serum. 5-Hydroxyindoleacetate metabolites of indoles and derivatives were also

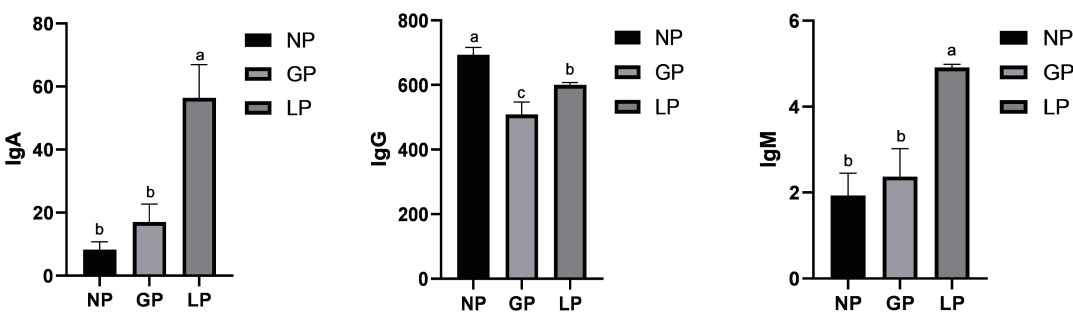

**FIG 8** Analysis of blood immune indices in different reproductive stages. NP, non-pregnancy; Gp, pregnancy; LP, lactation.

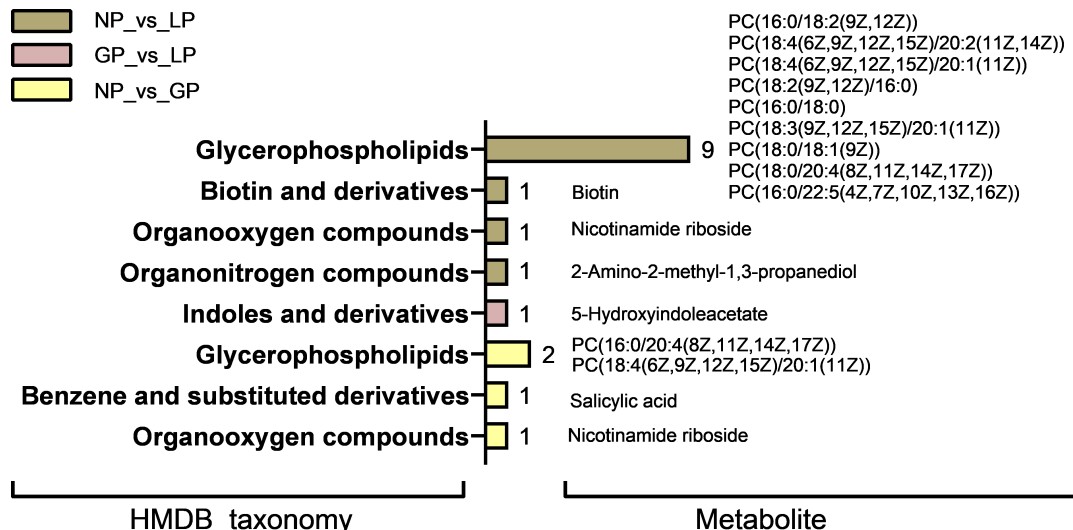

**FIG 9** HMDB functional classification of common differential metabolites. NP, non-pregnancy; Gp, pregnancy; LP, lactation.

found to be decreased in both the rumen and serum in lactation. In addition, nine differential metabolites were found in the non-pregnancy and lactation stages and annotated as glycerophospholipids, among which the contents of nicotinamide riboside and ethyl 9-decenoate were decreased in the rumen and serum during lactation.

## DISCUSSION

There are a series of changes in maternal physiological metabolism in different reproductive stages. Especially during pregnancy, the gradual increase in the size of the uterus, compression of the gastrointestinal tract, influence of peristalsis (16), and changes in the levels of some hormones can lead to changes in the structure of the gut microbiota (8). In this study, we found that the abundances of Firmicutes and *Prevotella*, which are related to nutrient digestion and absorption (17) and fiber degradation (18, 19), increased during pregnancy in Small-tailed Han sheep. During lactation, the abundances of Fibrobacteres, Spirochaetes, and *Fibrobacter* increased, and it has been found that these bacteria can produce VFAs through some carbohydrate degrading enzymes (20–22). VFA may be required during lactation to meet normal energy metabolism. Our further VFA measurement also found that the total VFA content was the highest during lactation, but the relationship between these microflora and the normal energy metabolism of lactating ewes still needs further study. In addition, ewes are prone to some metabolic diseases during pregnancy and lactation (13, 23). In this study, the abundances of *Helicobacter* and *Erysipelotrichaceae_UCG-009*, which are potentially disease-related marker microbes, were significantly reduced lactation (22, 24). Further functional analysis showed that carbohydrate metabolism, sugar biosynthesis, and metabolic function increased during pregnancy, indicating that Small-tailed Han sheep ewes have higher energy metabolism during pregnancy, which may play a certain role in the development of the fetus. Changes in the structure and function of the microbiota at different stages of reproduction lead to changes in its metabolites. We found that the bioactive compound sapindoside A, which is associated with anti-inflammation (25), is increased during pregnancy, while nicotinamide riboside, which is associated with the regulation of metabolic disorders (26), is decreased during pregnancy, suggesting that these metabolites may be involved in the regulation of metabolic homeostasis during pregnancy. During lactation, it was found that L-tryptophan (27), which is related to immune regulation, was significantly increased, thereby regulating the health of ewes during lactation and ensuring the healthy growth of lactating lambs. Further cluster analysis of these differential metabolites revealed that

phosphatidylcholine-related metabolites, which play a role in lipoprotein homeostasis and immune regulation (28, 29), were increased during pregnancy and lactation and reached the highest level during lactation. However, reproductive ewes are prone to metabolic disorders during pregnancy and lactation. This study found that β-cryptoxanthin, which is related to the regulation of metabolic disorders in the body (30), gradually decreased from the non-pregnancy, to the pregnancy and to the lactation stages. During pregnancy and lactation, the differential metabolites phylloquinone and menaquinone contents decreased. These metabolites play an important role in blood clotting, body metabolism (31), and bacterial energy production and bone health (32), indicating that these metabolites may play a role in pregnancy and lactation. In addition, the biosynthetic function of indole alkaloids is enriched during pregnancy, among which the increased metabolite harmaline can prevent the occurrence of inflammation and effectively protect maternal and fetal health (33). In lactation, differential metabolites were enriched in glycine, serine, and threonine metabolism, among which the increased L-tryptophan also played an important role in immune regulation (27), ensuring maternal health in lactation. In addition, energy metabolic homeostasis during pregnancy and lactation is important for maternal and offspring development. We found that the energy metabolite butyrate is highest during pregnancy and that butyrate and acetate can be interconverted (34). However, the acetic acid content increases during lactation, and some VFAs may reach the breast milk through blood. Studies have found that high concentrations of VFAs in breast milk can be transmitted to offspring (35) and promote the development of the offspring's immune system (36)(37). Furthermore, the rumen $NH_3$-N content increases during pregnancy, which is consistent with the results of Close et al. (38), and this $NH_3$-N provides a certain nitrogen source during pregnancy. However, are there any links between these metabolite changes and the microbiota? In this study, carnitine, which plays a key role in energy metabolism (39, 40), was found to be significantly positively correlated with *Ruminococcaceae*, suggesting that maternal energy metabolism may be regulated through microbial lipid metabolism during pregnancy (41). Menadione, which is related to energy production (32), is present at higher levels during pregnancy than during lactation and is significantly positively correlated with *Prevotella*, which is also highest during pregnancy. They may be involved in the regulation of energy metabolism during pregnancy, but this needs to be tested in animals. In addition, 2-undecanone has a significant positive correlation with *Anaerovibrio*, and studies have reported that 2-undecanone can prevent the occurrence of inflammation (42, 43), which is higher in the pregnancy stage than in the lactation stage, possibly to more effectively prevent the occurrence of inflammation in pregnant mothers to ensure the normal development of the fetus.

Metabolites are transported to various tissues and organs of the body through the blood to perform their functions. This study further identified the blood metabolites of Small-tailed Han sheep in different reproductive stages and found that inhibition of inflammation-associated 25-hydroxycholesterol (44, 45) is increased during pregnancy and may play a role during pregnancy, which requires further validation. Nicotinuric acid is a potential marker of metabolic syndrome (46) and was reduced during pregnancy in the present study, so it can be considered a potential metabolite for metabolic disorders during pregnancy. In addition, PC [22:4(7Z,10Z,13Z,16Z)/P-18:0] decreased during pregnancy and lactation, which was exactly the opposite of the previous metabolite content in rumen fluid. Studies have found that phosphatidylcholine plays a certain role in the regulation of lipoprotein homeostasis and immunity (28, 29). KEGG functional analysis showed that the function of serum metabolites was consistent with that of rumen fluid metabolites. The differential metabolites in non-pregnancy and lactation are enriched in ABC transporters, and ABC transporters participate in various physiological processes of different human tissues and regulate the development and function of different T-cell populations (47), thus regulating the immune status of the body. Among them, the increased metabolite biotin plays an important role in immune diseases (48),

while the content of L-glutamine, a metabolite related to antioxidant and anti-inflammatory effects (49), is decreased in lactation.

In ruminants, before and after delivery is one of the most critical stages in the reproductive cycle (50), and changes in the microbiota and its metabolites can affect the immune status of animals. In this study, the IgG content in lactation was significantly higher than that at other stages, while it was the lowest in the blood of sheep and goats from 15 days before delivery as reported, and the results were consistent with a gradual increase during the postpartum stage (51). The reason is that most of the IgG in the blood is transferred to the colostrum (52), thereby boosting the immunity of the lamb through the colostrum. The IgM content during lactation was significantly higher, indicating that the high-active IgM content of Small-tailed Han sheep in lactation played a role in removing pathogens (53) and improving the immunity of the body, which was consistent with the fact that the immunity of lactating lambs was strong. The IgA content in the blood is the highest in lactation, and studies have shown that acetic acid can induce the generation of IgA by regulating the interaction between intestinal epithelial cells and immune cells (54) to maintain a higher immune level in lactation. Therefore, different types of serum immunoglobulins play different immune functions in different reproductive stages of Small-tailed Han sheep and have interaction effects with intestinal immune cells, but this requires further verification. Further comparative analysis of the differential microbial metabolites and serum metabolites revealed that there were common differential metabolites, nicotinamide riboside was decreased in the rumen and serum during pregnancy and lactation, and it played a role in metabolic disorders (26), so it could be assumed to be a potential marker of metabolic disorders, but further verification is needed. In the comparison between pregnancy and lactation, it was found that 5-hydroxyindoleacetate decreased in lactation, while 5-hydroxyindoleacetate could enhance the accumulation of lipids in preadipocytes (55). Therefore, more 5-hydroxyindoleacetate may be needed in lactation to maintain the fat synthesis and milk fat synthesis of ewes to meet the nutritional requirements of lactating lambs, which provides an idea for future research.

## Conclusion

This study elucidated the succession of the rumen microbiota, its metabolites, and host metabolites in Small-tailed Han sheep during non-pregnant, pregnant, and lactating periods. The abundances of Firmicutes and *Prevotella*, which are related to energy metabolism, increased during pregnancy and the abundance of Fibrobacter increased during lactation. In addition, increased carbohydrate metabolism, sugar biosynthesis, and metabolic functions of the microbiota were observed during pregnancy. In addition, some immune-related metabolites were autonomously regulated by the body during pregnancy and lactation and were enriched in the indole alkaloid biosynthesis and glycine, serine, and threonine metabolism pathways. A high content of $NH_3$-N is found during pregnancy to ensure the growth and development needs of the fetus. In addition, blood metabolites related to anti-inflammation and disease resistance were increased during pregnancy and enriched in arachidonic acid metabolism and vitamin B6 metabolism during pregnancy and laceration. Participating in the regulation of the body's immune level, IgA and IgM significantly increased in lactation, to ensure the health of the mother and lamb. This study provides a reference for the health management of ruminants during the non-pregnancy, pregnancy, and lactation stages.

## MATERIALS AND METHODS

### Experimental design and sample collection

In this study, 15 ewes, each 3 years old, from the Small-tailed Han sheep breed with a similar body weight of 60.0 ± 2.5 kg and good health were selected as the study objects, and the whole reproductive cycle (non-pregnancy, pregnancy, and lactation)

was tracked. All sheep received uniform feed formula and feeding and management conditions during the experimental stages and were free to eat and drink. Blood and rumen content samples were collected during the non-pregnancy stages (15 d before insemination), pregnancy stages (120 d after pregnancy), and lactation stages (40 d after delivery). Three milliliters of whole blood were collected from the jugular vein and centrifuged at 3,000 rpm for 15 min. The serum was separated, transferred to a frozen storage tube, placed in a liquid nitrogen tank, and brought back to the laboratory for storage at −80°C for the determination of immune indices and the blood metabolome. In addition, 50 mL of rumen fluid was extracted using a rumen fluid sampler with a gastric tube. It was immediately filtered with four layers of sterile gauze, part of which was stored in a 2-mL sterile frozen tube, numbered and placed in liquid nitrogen, and brought back to the laboratory for storage at −80°C for microbial DNA extraction, rumen microbial 16S rRNA sequencing, and microbial metabolite identification. The remaining rumen fluid was divided into frozen tubes and stored at −20°C for the determination of the VFA content and $NH_3$-N content.

## Determination of blood immune indices and rumen VFA and $NH_3$-N contents

The enzyme linked immunosorbent assay (ELISA) kit for the determination of blood immune indices was purchased from Nanjing Jiancheng Bioengineering Institute and performed in strict accordance with the operation steps of the instructions. The contents of IgA, IgG, and IgM in blood were detected and analyzed with Thermo 3020 enzyme-labeled instrument. The rumen VFA content of Small-tailed Han sheep at different reproductive stages was determined by GC-2010 Plus gas chromatograph. The internal standard method was used with 2-ethylbutyric acid as the internal standard. Chromatography was performed on an Analytical Technology free fatty acid phase (AT-FFAP) (50 m × 0.32 mm × 0.25 µm) capillary column. The temperature of the column was kept at 60°C for 1 min, then increased to 115°C at 5°C/min without reservation, and then increased to 180°C at 15°C/min. The temperature of the detector was 260°C, and the temperature of the inlet was 250°C. The $NH_3$-N content was determined according to the method of Chaney et al. (56).

## 16s rRNA sequencing of the rumen microbiota

Rumen microbiome DNA was extracted using a bacterial DNA extraction kit MN NucleoSpin 96 Soi (Omega, Shanghai, China). The conserved region of nucleotides encoding bacterial ribosomal RNA is mainly the 16S region. PCR was used to amplify the V3–V4 regions of the highly variable region of the 16S rRNA gene using the forward primer 338F: 5′-ACTCCTACGGGAGGCAGCA-3′ and reverse primer 806R: 5′-GGAC-TACHVGGGTWTCTAAT-3′. The rumen microbiota was sequenced by a two-step database construction method, and the amplified products were sequenced by an Illumina MiSeq 2500 (Illumina, San Diego, CA, USA) platform. Reads of the original sequencing data were double-end spliced (FLASH, version 1.2.7) to obtain the original tags data, and Trimmomatic (version 0.33) was used for quality filtering to obtain high-quality Tags data. Chimera sequences were identified and removed (UCHIME, version 4.2) to obtain the final valid data. Usearch (version 10.0) software was used to perform cluster analysis of the high-quality valid data, and the 2013 Greengenes (version 13.8) ribosome database at a similarity level of 97% was used to obtain the OTUs (57). Alpha diversity analysis was performed on the OTU analysis results by Mothur (version 1.30) to study species diversity within a single sample and draw dilution curves (58). The PCoA diagram of the corresponding distance samples was obtained according to the distance matrix algorithm. LEfSe was used to analyze biomarkers with statistical differences in different reproductive stages (59). The Silva (Bacteria) taxonomic database was used for OTU classification labeling analysis, and microbial species composition at different taxonomic levels (phylum, class, order, family, genus, and species) was obtained. Furthermore, PICRUSt software was used to predict the KEGG functions and COG of the 16S rRNA

sequencing data, and the differences and changes in the rumen microbiota in metabolic pathways in different reproductive stages were analyzed.

## Metabolite determination

Serum samples and rumen fluid samples at different stages were analyzed by liquid chromatography-mass spectrometry. After the samples were thawed at room temperature, 100-µL samples were taken each time, and 500 µL of extraction solution containing internal standard (1,000:2) was added (methanol-acetonitrile volume ratio = 1:1, internal standard concentration 2 mg/L) and vortexed for 30 s. Then, the samples were ultrasounded in an ice water bath for 10 min, left at −20℃ for 1 h, and centrifuged at 4℃ for 15 min (12,000 rpm). Next, 500 µL of supernatant was removed from the Eppendorf (EP) tube, and the extract was dried in a vacuum concentrator. Then, 150 µL of extract solution (acetonitrile-water volume ratio: 1:1) was added to the dried metabolites for redissolution, continued to swirl for 30 s, ultrasonicated in an ice water bath for 10 min, and centrifuged at 4℃ for 15 min (12,000 rpm). Finally, 120 µL of supernatant was taken out into 2-mL injection bottle, and each sample was mixed with 10 µL to form QC samples for machine detection. The LMS system for metabolomics analysis consisted of a Waters Acquity I-Class PLUS ultra-high-performance liquid phase tandem Waters Xevo G2-XS QToF high-resolution mass spectrometer. An Acquity UPLC HSS T3 column (1.8 µm 2.1 * 100 mm) purchased from Waters was used. The samples were eluted in positive ion mode (ESI+) and negative ion mode (ESI−) with a mobile phase consisting of water and 5% acetonitrile, 0.1% formic acid as solvent A, and 0.1% acetonitrile and 0.1% formic acid as solvent B at flow rates of 0.35 mL/min and 400 µL/min. The subsequent mobile phase (A:B) elution gradient was as follows: 98%:2% during 0–0.25 min, 2%:98% during 10.0–13.0 min, 98%:2% during 13.1–15.0 min, with an ion source temperature of 150℃ and a desolvent temperature of 500℃. The flow rates of the backflow and desolvent gas were 50 L/h and 800 L/h, respectively.

## Data analysis

According to the method of McHardy et al. (60), the microbiome and metabolome were jointly analyzed, and PCoA was used to reduce the dimensions of the microbiome (genus level) and metabolome, respectively. First, the distance matrix was calculated by the quantitative matrix of microbiota and metabolites. The distance algorithm of the microbiome was Bayesian distance, the distance algorithm of metabolome was Euclidean distance, and PCoA was used to sort the distance. The coordinates of the characteristic axes in the PCoA results of the microbiome and metabolome were extracted, and Procrustes analysis was performed to compare the similarity and variation between the microbiome and metabolome. Metabolic data were dimensionally reduced by weighted gene co-expression network analysis, and metabolites were divided into different metabolite clusters, with the expression of metabolite clusters represented by the median content in the same cluster. Pearson correlation analysis was performed with the microbiota, heatmaps were drawn, and the correlation analysis results were screened. The screening condition was the $P$ value, and the standard was CCP <0.05. Then, the frequency of occurrence of metabolite clusters/microbiota was counted, and the correlation result table of metabolite clusters/microbiota with the top 30 frequencies was drawn to create a chord diagram. IBM SPSS Statistics 26.0 software (SPSS, Inc., Chicago, Illinois, USA) was used for data analysis. Single-factor analysis of variance was used to analyze the contents of VFAs and $NH_3$-N and the immune indices in the rumen contents of Small-tailed Han sheep at different reproductive stages. The results were all represented as "mean ± standard deviation," and the correlation analysis was conducted by the Spearman correlation test, with a statistical significance level of $P < 0.05$.

## ACKNOWLEDGMENTS

Thanks to all participants for their advice and support of this study.

This research was funded by the National Natural Science Foundation of China (32260820), Gansu HOME Program Characteristic Demonstration Project (GSHZSF2023-01), Discipline Team Project of Gansu Agricultural University (GAU‐XKTD‐2022‐21), Gansu Province science and technology plan project (23JRRA1447), and Gansu Agricultural University Youth Mentor Support Fund project (GAU‐QDFC‐2022‐06).

Y.S. and X.L. designed the study. X.P., Y.H., J.W., S.Z., P.S., F.W., Z.X., X.C., and W.Y. performed the experiments and collected samples. Y.S. analyzed the data and wrote the manuscript. All authors read and approved the final manuscript.

## AUTHOR AFFILIATIONS

¹College of Animal Science and Technology/Gansu Key Laboratory of Herbivorous Animal Biotechnology, Gansu Agricultural University, Lanzhou, China
²School of Fundamental Sciences, Massey University, Palmerston North, New Zealand

## AUTHOR ORCIDs

Yuzhu Sha  http://orcid.org/0000-0003-0023-8628
Xiu Liu  http://orcid.org/0000-0001-6057-2551

## FUNDING

| Funder | Grant(s) | Author(s) |
| --- | --- | --- |
| National Natural Science Foundation of China | 32260820 | Xiu Liu |
| Gansu HOME Program Characteristic Demonstration Project | GSHZSF2023-01 | Xiu Liu |
| Discipline Team Project of Gansu Agricultural University | GAU‐XKTD‐2022‐21 | Xiu Liu |
| Gansu Province Science and technology plan project | 23JRRA1447 | Yuzhu Sha |
| Gansu Agricultural University Youth Mentor Support Fund project | GAU‐QDFC‐2022‐06 | Xiu Liu |

## AUTHOR CONTRIBUTIONS

Yuzhu Sha, Conceptualization, Formal analysis, Methodology, Writing – original draft, Writing – review and editing | Xiu Liu, Conceptualization, Data curation, Funding acquisition, Investigation, Writing – review and editing | Xiaoning Pu, Investigation, Resources | Yanyu He, Formal analysis, Methodology, Writing – review and editing | Jiqing Wang, Software | Shengguo Zhao, Investigation, Software | Pengyang Shao, Resources, Software | Fanxiong Wang, Resources | Zhuanhui Xie, Investigation, Resources | Xiaowei Chen, Investigation | Wenxin Yang, Resources

## DATA AVAILABILITY

The data sets presented in this study can be found in the NCBI Sequence Read Archive (SRA) under accession numbers SRR24962442 to SRR24962456 (PRJNA984421).

## ETHICS APPROVAL

The animals involved in the research are subject to the Regulations of the Ministry of Science and Technology, PRC, on the Administration of Laboratory Animal Affairs (PRC; Revised in June 2004). The sample collection program has been approved by the Animal Husbandry Specialty Committee of Gansu Agricultural University (Approval No. GAU-LC-2020-27).

## ADDITIONAL FILES

The following material is available online.

## Supplemental Material

**Supplemental material (Spectrum02867-23-s0001.docx).** Tables S1 to S3 and Fig. S1 to S7.

## Open Peer Review

**PEER REVIEW HISTORY (review-history.pdf).** An accounting of the reviewer comments and feedback.

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
