## [Reviewer comments · Microbiology Spectrum]

Microbiology Spectrum

Characterizing the dynamics of the rumen microbiota, its metabolites, and blood metabolites across reproductive stages in Small-tailed Han sheep

Yuzhu Sha, Xiu Liu, Xiaoning Pu, Yanyu He, Jiqing Wang, Shengguo Zhao, Pengyang Shao, Fanxiong Wang, Zhuanhui Xie, Xiaowei Chen, and Wenxin Yang

Corresponding Author(s): Xiu Liu, Gansu Agricultural University

Review Timeline:

Submission Date:	July 17, 2023
Editorial Decision:	September 14, 2023
Revision Received:	October 10, 2023
Accepted:	October 23, 2023

Editor: John Atack

Reviewer(s): The reviewers have opted to remain anonymous.

Transaction Report:

DOI: <https://doi.org/10.1128/spectrum.02867-23>

September 14, 2023

Prof. Xiu Liu
Gansu Agricultural University
College of Animal Science and Technology
Lanzhou, Gansu
China

Re: Spectrum02867-23 (Characterizing the dynamics of rumen microbiota, their metabolites, and blood metabolites across reproductive stages in Small Tail Han sheep)

Dear Prof. Xiu Liu:

Link Not Available

Sincerely,

John Atack

Journals Department
Reviewer comments:

Reviewer #1 (Comments for the Author):

The authors investigated associations between the rumen microbiota, microbial metabolites (predicted using PICRUSt) and blood metabolites at three reproductive stages (non-pregnant, pregnant, lactating) in a group of Small Tail Han sheep.

This work is scientifically sound, but I feel that this manuscript needs to go through several further rounds of revision with input from senior authors before it will be acceptable for publication. At present, this reads more like a student manuscript and is not of sufficient quality for publication. The manuscript would also benefit from input from a native English speaker as some of the content is unclear.

Some key points:

- Many of the figures need to be revised and made larger or moved to the supplementary material
- Abbreviations needed to be explained and used consistently throughout the manuscript
- The discussion could be abbreviated as much of it is a repetition of the results

I had begun making some edits to the Word doc (see file), but this is probably something that the authors should complete.

Reviewer #3 (Comments for the Author):

This manuscript evaluated the ewe gut microbiota and metabolome during non-pregnancy, pregnancy, and lactation. The authors evaluated the correlation between microbiota and metabolites during these three physiological stages, and identified some important microorganisms and metabolites, which may have some health indication for sheep pregnancy and lactation. In general, this article provided some decent information for the microbiome and metabolome of ewe gut during these three physiological stages.

However, the authors seem to exaggerate the effects of certain bacteria and metabolites in the ewe gut during pregnancy/lactation and confuse the causation and correlation effects. For example, the conclusion drawn by the authors are not convincing (Line 498-499). The roles of *Fibrobacter* during pregnancy is not proved in this study, since no in vitro or in vivo experiments were performed to evaluate the effects of these bacteria. The authors performed only amplicon metagenomics and MS-based metabolomics, and the evidence is lacking for this conclusion. Another example in the discussion section (Line 325-327), Firmicutes carry genes related to energy metabolism, and their abundance increased during pregnancy. But this information alone does not provide basis that firmicutes help host digest and absorb nutrients, which is a general and bold statement without evidence. Similar statements can be found throughout the discussion section (Line 328-330, 350, 376-378). The argument on the association of blood antibody levels (IgA, IgG, and IgM) and the gut microbiome is weak. If the three reproductive stages have different levels of blood antibodies and different gut microbiota, this information alone does not provide association between blood antibody and gut microbiota.

Additionally, the authors seem to lack the general microbiology knowledge, and they used the sequencing analyses software denoted names, instead of the correct genus names throughout the article. Please make sure to remove any suffix from sequencing analysis. For example, Line 146-147, "*Prevotella_1*", "*Rikenellaceae_RC9_gut_group*", so on and so forth. Please confirm the species of "SP3-e08" (Line 252).

The authors should provide higher quality figures for Fig 5, 6, 7, 9. It is important to make figures self-explanatory, and the figure captions should be more informative. Please make sure to explain relevant abbreviations in the figure captions as well. For example, explain abbreviation "Gp", "Lp", "Np" in figure captions.

The authors are suggested to review the manuscript carefully and make sure to use capitalization and italics properly.

Here are some additional specific suggestions for the authors:

Line 130: typo "OTUs"

Line 199, 282: "Venn diagram"

Line 339: It is not correct to call *Helicobacter* and *Erysipelotrichaceae* pathogenic. Some species may be pathogenic to ewes, but not all genus/family are pathogenic.

Line 376-378: This indication is not valid based on the information provided by the authors. The "need" for vitamin supplements and its importance for pregnancy/lactation requires additional references or experiments.

Line 418-420: "ensure the health" is too strong a claim. Provide either provide literature for these claims or change the language.

Line 631: change "gate" to "phylum"

Figure 1 caption: missing information. What is the meaning of different color and shape of objects in Fig 1C?

Fig 3 is not informative, and the authors did not explain the meaning of this figure properly in the caption. What is the LDA scores of other OTUs, and why only show the OTUs from group "Gp"? What about the other two groups?

Staff Comments:

Preparing Revision Guidelines

- Point-by-point responses to the issues raised by the reviewers in a file named "Response to Reviewers," NOT IN YOUR

COVER LETTER.

- Upload a compare copy of the manuscript (without figures) as a "Marked-Up Manuscript" file.
- Each figure must be uploaded as a separate file, and any multipanel figures must be assembled into one file.
- Manuscript: A .DOC version of the revised manuscript
- Figures: Editable, high-resolution, individual figure files are required at revision, TIFF or EPS files are preferred

Please return the manuscript within 60 days; if you cannot complete the modification within this time period, please contact me. If you do not wish to modify the manuscript and prefer to submit it to another journal, please notify me of your decision immediately so that the manuscript may be formally withdrawn from consideration by Microbiology Spectrum.

This manuscript evaluated the ewe gut microbiota and metabolome during non-pregnancy, pregnancy, and lactation. The authors evaluated the correlation between microbiota and metabolites during these three physiological stages, and identified some important microorganisms and metabolites, which may have some health indication for sheep pregnancy and lactation.

In general, this article provided some decent information for the microbiome and metabolome of ewe gut during these three physiological stages.

However, the authors seem to exaggerate the effects of certain bacteria and metabolites in the ewe gut during pregnancy/lactation and confuse the causation and correlation effects. For example, the conclusion drawn by the authors are not convincing (Line 498-499). The roles of *Fibrobacter* during pregnancy is not proved in this study, since no in vitro or in vivo experiments were performed to evaluate the effects of these bacteria. The authors performed only amplicon metagenomics and MS-based metabolomics, and the evidence is lacking for this conclusion. Another example in the discussion section (Line 325-327), Firmicutes carry genes related to energy metabolism, and their abundance increased during pregnancy. But this information alone does not provide basis that firmicutes help host digest and absorb nutrients, which is a general and bold statement without evidence. Similar statements can be found throughout the discussion section (Line 328-330, 350, 376-378). The argument on the association of blood antibody levels (IgA, IgG, and IgM) and the gut microbiome is weak. If the three reproductive stages have different levels of blood antibodies and different gut microbiota, this information alone does not provide association between blood antibody and gut microbiota.

Additionally, the authors seem to lack the general microbiology knowledge, and they used the sequencing analyses software denoted names, instead of the correct genus names throughout the article. Please make sure to remove any suffix from sequencing analysis. For example, Line 146-147, “*Prevotella_1*”, “*Rikenellaceae_RC9_gut_group*”, so on and so forth. Please confirm the species of “SP3-e08” (Line 252).

The authors should provide higher quality figures for Fig 5, 6,7, 9. It is important to make figures self-explanatory, and the figure captions should be more informative. Please make sure to explain relevant abbreviations in the figure captions as well. For example, explain abbreviation “Gp”, “Lp”, “Np” in figure captions.

The authors are suggested to review the manuscript carefully and make sure to use capitalization and italics properly.

Here are some additional specific suggestions for the authors:

Line 130: typo “OTUs”

Line 199, 282: “Venn diagram”

Line 339: It is not correct to call *Helicobacter* and *Erysipelotrichaceae* pathogenic. Some species may be pathogenic to ewes, but not all genus/family are pathogenic.

Line 376-378: This indication is not valid based on the information provided by the authors. The “need” for vitamin supplements and its importance for pregnancy/lactation requires additional references or experiments.

Line 418-420: “ensure the health” is too strong a claim. Provide either provide literature for these claims or change the language.

Line 631: change “gate” to “phylum”

Figure 1 caption: missing information. What is the meaning of different color and shape of objects in Fig 1C?

Fig 3 is not informative, and the authors did not explain the meaning of this figure properly in the caption. What is the LDA scores of other OTUs, and why only show the OTUs from group “Gp”? What about the other two groups?

Response to Reviewers:

Reviewer #1 (Comments for the Author):

The authors investigated associations between the rumen microbiota, microbial metabolites (predicted using PICRUST) and blood metabolites at three reproductive stages (non-pregnant, pregnant, lactating) in a group of Small Tail Han sheep.

Question: This work is scientifically sound, but I feel that this manuscript needs to go through several further rounds of revision with input from senior authors before it will be acceptable for publication. At present, this reads more like a student manuscript and is not of sufficient quality for publication. The manuscript would also benefit from input from a native English speaker as some of the content is unclear.

Reply: We are very grateful to the reviewers for their recognition of our research, and for spending their precious time to carefully review our manuscript and put forward valuable suggestions, which have greatly helped to improve the quality of our research content. There are indeed many problems in our first draft. Thanks to the reviewers for raising these problems, we have revised these problems. In addition, the grammar of the article has also been checked and modified by native English speakers. The specific modification reply is as follows:

Some key points:

- Many of the figures need to be revised and made larger or moved to the supplementary material

Reply: We have modified all the figures, enlarged the text and other information in the figures, and put some figures that are not very important into the supplementary materials. In addition, according to the editor's request, we removed the pictures from the manuscript and uploaded them as separate files.

- Abbreviations needed to be explained and used consistently throughout the manuscript

Reply: Thank you very much for the question raised by the reviewer. We have explained the full name of all the acronyms and kept the whole text consistent.

- The discussion could be abbreviated as much of it is a repetition of the results

Reply: We have made a lot of changes to the discussion section and simplified the description of some sentences. Some duplicates in the results section were removed.

I had begun making some edits to the Word doc (see file), but this is probably something that the authors should complete.

Reply: Thank the reviewers for taking the time to review our manuscript and making valuable suggestions. We have revised the whole article according to the reviewer's comments.

Reviewer #3 (Comments for the Author):

This manuscript evaluated the ewe gut microbiota and metabolome during non-pregnancy, pregnancy, and lactation. The authors evaluated the correlation between microbiota and metabolites during these three physiological stages, and identified some important microorganisms and metabolites, which may have some health indication for sheep pregnancy and lactation.

In general, this article provided some decent information for the microbiome and metabolome of ewe gut during these three physiological stages.

Reply: Thanks to the reviewers for their recognition of this study and for taking their valuable time to review our manuscript and make some helpful suggestions. According to your valuable suggestions, we have made some changes to the full text. The following is the revised reply:

Question: However, the authors seem to exaggerate the effects of certain bacteria and metabolites in the ewe gut during pregnancy/lactation and confuse the causation and correlation effects. For example, the conclusion drawn by the authors are not convincing (Line 498-499). The roles of *Fibrobacter* during pregnancy is not proved in this study, since no in vitro or in vivo experiments were performed to evaluate the effects of these bacteria. The authors performed only amplicon metagenomics and MS-based metabolomics, and the evidence is lacking for this conclusion.

Another example in the discussion section (Line 325-327), Firmicutes carry genes related to energy metabolism, and their abundance increased during pregnancy. But this information alone does not provide basis that firmicutes help host digest and absorb nutrients, which is a general and bold statement without evidence. Similar statements can be found throughout the discussion section (Line 328-330, 350, 376-378).

Reply: Thank you very much for this question raised by the reviewer. Due to our inappropriate description in the previous draft, we did exaggerate the function of metabolites in pregnancy and lactation. Therefore, we have made some changes to the discussion section, paying more attention to the rigor of the language. In this study, we studied rumen microbiota and their metabolites of ewes at different reproductive stages to explore whether there were significant changes in some microbiota and metabolites at different reproductive stages, screened out some different microbiota and their metabolites. According to the previous research progress and the enriched functions in this study, the possible roles of them in the breeding stage of ewe were preliminarily discussed. These findings will provide directions for targeting microorganisms/metabolites at the later stages of ewe reproduction. However, we have not been able to do the specific function of these differential microbiota/metabolites in the reproductive stage, we are not clear, so further verification is needed in the future. Therefore, the reviewer's suggestions are of great help to us, and we have modified some sentences in the discussion part to ensure their rigor. For specific modifications, see L288-293, L305-309, L319-324, L341-345, and elsewhere in the discussion section.

Question: The argument on the association of blood antibody levels (IgA, IgG, and IgM) and the gut microbiome is weak. If the three reproductive stages have different levels of blood antibodies and different gut microbiota, this information alone does not provide association between blood antibody and gut microbiota.

Reply: Thanks to the reviewer for raising this question, we have revised the manuscript to

some extent. This study analyzed rumen microbiota and their metabolites at different reproductive stages, explored the changes of rumen microbiota and their metabolites at different reproductive stages, and preliminarily explored the effects of these changes on ewes according to previous research progress. And according to the function of microbe and metabolite enrichment in this study (see L300-303, L320-331, L361-369, et al), we found that it plays a role in immunity and energy metabolism. Therefore, further analysis of the non-targeted metabolome of the blood was performed to find out whether the host metabolites played a role at different stages, and immunoglobulins were measured to reveal the immune status of ewes at different breeding stages. We did not directly analyze the association between microbiota and blood, and it may be that the relationship between them is mainly influenced by some metabolites, Therefore, this study conducted a comparative analysis of microbial metabolites and blood metabolites (see L387-396), and indeed found that there are common metabolites, and according to previous studies, these metabolites are associated with disease and energy, but this needs to be further verified in animal models. Thank you for asking this question, which has given us a great direction for the subsequent test design.

Question: Additionally, the authors seem to lack the general microbiology knowledge, and they used the sequencing analyses software denoted names, instead of the correct genus names throughout the article. Please make sure to remove any suffix from sequencing analysis. For example, Line 146-147, "Prevotella_1", "Rikenellaceae_RC9_gut_group", so on and so forth. Please confirm the species of "SP3-e08" (Line 252).

Reply: Thanks to the reviewer for raising the problem of microorganism naming, we have deleted the suffix. In addition, the microbe was named in reference to the Silva138 database, which was named after researchers who had actually studied the strain. SP3-e08 belongs to a genus of bacteria in Rikenellaceae (<https://www.arb-silva.de/browser/ssu-121/AB331468/>). This microbe was also found in other articles, so the name of this study is appropriate.

[1] Donielle Pannell, Brou Kouakou, Thomas H. Terrill, et al. Adding dried distillers grains with solubles influences the rumen microbiome of meat goats fed lespedeza or alfalfa-based diets, *Small Ruminant Research*, 214,2022,106747, <https://doi.org/10.1016/j.smallrumres.2022.106747>.

[2] Li Y, Yang Y, Chai S, et al. Ruminal Fluid Transplantation Accelerates Rumen Microbial Remodeling and Improves Feed Efficiency in Yaks. *Microorganisms*. 2023 Jul 31;11(8):1964. doi: 10.3390/microorganisms11081964.

Question: The authors should provide higher quality figures for Fig 5, 6,7, 9. It is important to make figures self-explanatory, and the figure captions should be more informative. Please make sure to explain relevant abbreviations in the figure captions as well. For example, explain abbreviation "Gp", "Lp", "Np" in figure captions.

Reply: The figure was modified and a high quality figure file was uploaded. Some acronyms are explained in the notes.

The authors are suggested to review the manuscript carefully and make sure to use capitalization and italics properly.

Reply: Thanks for the reviewer's questions, we have checked the uppercase and italics of the full text, and made modifications to maintain the consistency of the full text. Genus level microbes and genes are written in italics.

Here are some additional specific suggestions for the authors:

Line 130: typo "OTUs"

Reply: Have been modified

Line 199, 282: "Venn diagram"

Reply: Have been modified

Line 339: It is not correct to call Helicobacter and Erysipelotrichaceae pathogenic. Some species may be pathogenic to ewes, but not all genus/family are pathogenic.

Reply: Thank the reviewer for asking this question. It is true that not all genera/families are pathogenic, and we have revised our description to bring more rigor to the discussion.

Line 376-378: This indication is not valid based on the information provided by the authors. The "need" for vitamin supplements and its importance for pregnancy/lactation requires additional references or experiments.

Reply: Thank you very much for the question raised by the reviewer. We are also aware of the error of this description, we do not have any evidence to suggest the need for vitamin supplements, our conclusions are too absolute. Therefore, we modified this description and found that it is mainly the enrichment of these differential metabolites during gestation and lactation, which may play a certain role, but further verification is still needed.

Line 418-420: "ensure the health" is too strong a claim. Provide either provide literature for these claims or change the language.

Reply: Thank you for the question raised by the reviewer. We have made modifications. the phrase "ensure the health" has been deleted.

Line 631: change "gate" to "phylum"

Reply: Have been modified

Figure 1 caption: missing information. What is the meaning of different color and shape of objects in Fig 1C?

Reply: Have been modified

Fig 3 is not informative, and the authors did not explain the meaning of this figure properly in the caption. What is the LDA scores of other OTUs, and why only show the OTUs from group "Gp"? What about the other two groups?

Reply: Thank you for your question. We have changed the title to provide some information. LEfSe (Line Discriminant Analysis (LDA) Effect Size) analysis is used to find the Biomarker with statistical difference between different groups. Through comparative analysis of the three periods (LEfSe analysis), only three different biomarkers were found in the Gp period, and no other groups were found.

Re: Spectrum02867-23R1 (Characterizing the dynamics of the rumen microbiota, its metabolites, and blood metabolites across reproductive stages in Small-tailed Han sheep)

Dear Prof. Xiu Liu:

Your manuscript has been accepted, and I am forwarding it to the ASM production staff for publication. Your paper will first be checked to make sure all elements meet the technical requirements. ASM staff will contact you if anything needs to be revised before copyediting and production can begin. Otherwise, you will be notified when your proofs are ready to be viewed.

Sincerely,
John Atack
Editor
Microbiology Spectrum